# Cellular and Immunological Analysis of 2D2/Th Hybrid Mice Prone to Experimental Autoimmune Encephalomyelitis in Comparison with 2D2 and Th Lines

**DOI:** 10.3390/ijms25189900

**Published:** 2024-09-13

**Authors:** Kseniya S. Aulova, Andrey E. Urusov, Aleksander D. Chernyak, Ludmila B. Toporkova, Galina S. Chicherina, Valentina N. Buneva, Irina A. Orlovskaya, Georgy A. Nevinsky

**Affiliations:** 1Institute of Chemical Biology and Fundamental Medicine of the Siberian Division of RAS, Lavrentiev Ave. 8, Novosibirsk 630090, Russia; 2Institute of Systematics and Ecology of Animals of the Siberian Division of the RAS, Novosibirsk 630091, Russia; 3Institute of Clinical Immunology, Siberian Branch of the Russian Academy of Sciences, Novosibirsk 630090, Russia

**Keywords:** EAE mouse models, multiple sclerosis, hybrid 2D2/Th mice, stem cells, cell differentiation profiles, catalytic IgGs, hydrolysis of DNA and proteins

## Abstract

Previously, we described the mechanisms of development of autoimmune encephalomyelitis (EAE) in 3-month-old C57BL/6, Th, and 2D2 mice. The faster and more profound spontaneous development of EAE with the achievement of deeper pathology occurs in hybrid 2D2/Th mice. Here, the cellular and immunological analysis of EAE development in 2D2/Th mice was carried out. In Th, 2D2, and 2D2/Th mice, the development of EAE is associated with a change in the differentiation profile of hemopoietic bone marrow stem cells, which, in 2D2/Th, differs significantly from 2D2 and Th mice. Hybrid 2D2/Th mice demonstrate a significant difference in these changes in all strains of mice, leading to the production of antibodies with catalytic activities, known as abzymes, against self-antigens: myelin oligodendrocyte glycoprotein (MOG), DNA, myelin basic protein (MBP), and five histones (H1–H4) hydrolyze these antigens. There is also the proliferation of B and T lymphocytes in different organs (blood, bone marrow, thymus, spleen, lymph nodes). The patterns of changes in the concentration of antibodies and the relative activity of abzymes during the spontaneous development of EAE in the hydrolysis of these immunogens are significantly or radically different for the three lines of mice: Th, 2D2, and 2D2/Th. Several factors may play an essential role in the acceleration of EAE in 2D2/Th mice. The treatment of mice with MOG accelerates the development of EAE pathology. In the initial period of EAE development, the concentration of anti-MOG antibodies in 2D2/Th is significantly higher than in Th (29.1-fold) and 2D2 (11.7-fold). As shown earlier, antibodies with DNase activity penetrate cellular and nuclear membranes and activate cell apoptosis, stimulating autoimmune processes. In the initial period of EAE development, the concentration of anti-DNA antibodies in 2D2/Th hybrids is higher than in Th (4.6-fold) and 2D2 (25.7-fold); only 2D2/Th mice exhibited a very strong 10.6-fold increase in the DNase activity of IgGs during the development of EAE. Free histones in the blood are cytotoxic and stimulate the development of autoimmune diseases. Only in 2D2/Th mice, during different periods of EAE development, was a sharp increase in the anti-antibody activity in the hydrolysis of some histones observed.

## 1. Introduction

The literature describes artificial catalytically active antibodies known as abzymes (ABZs) that act against stable analogs of transition states of different chemical reactions and their use [1,2,3,4,5,6,7,8]. Natural abzymes from biological liquids of mammals are well described in [9,10,11,12,13,14,15,16,17,18,19,20,21,22,23,24,25,26]. Natural Abs-abzymes that split different proteins, peptides, oligosaccharides, DNAs, nucleotides, and RNAs were found in different biological liquids of patients with several autoimmune pathologies (AIPs) [18,19,20,21,22,23,24,25,26]. Conditionally healthy humans usually lack ABZs [24,25,26]. However, auto-ABZs with shallow enzymatic activities that degraded polysaccharides [18], thyroglobulin [19,20], and vasoactive neuropeptide [21] were found in the blood of some contingently healthy volunteers. The blood plasma of systemic lupus erythematosus (SLE) and multiple sclerosis (MS) patients usually contains abzymes that hydrolyze oligosaccharides [18], myelin basic protein (MBP), DNAs, RNAs, and histones ([24,25,26] and references therein).

AIPs were first assumed to originate from very specific defects in bone marrow (BM) hematopoietic stem cells (BM-HSCs) [27]. This assumption was later confirmed in several studies of mouse models predisposed to the development of SLE [28] and experimental autoimmune encephalitis (EAE) [29,30,31,32]. Several models of mouse EAE are known, and they mimic human MS-specific facets well. C57BL/6 mice show the T and B responses of lymphocytes against very different antigens [33,34]. The second EAE-prone mouse model is the Th line of mice, which have a particular myelin-specific T cell receptor (TCR) [35]. This was obtained by crossing with knock-in mice with a myelin-specific immunoglobulin heavy chain [35]. The 2D2 model of EAE-prone mice is characterized by the expression of a MOG-specific T cell receptor [36,37]. C57BL/6 [29,30,31], Th [31], and 2D2 [32] mice were used to study the possible mechanisms in untreated mice and the antigen-accelerated development of EAE pathology. Some typical indicators of EAE pathology development (different clinical, neurological, and histological evidence, including optic neuritis) appeared, as described earlier [33,34,35,36,37], relatively early, only several months after these mice were immunized with MOG. Th, 2D2, and C57BL/6 mice demonstrated a relatively slow spontaneous and significantly faster MOG-induced development of EAE [29,30,31,32]. In contrast to previous studies [33,34,35,36,37], the initial stages of spontaneous and antigen-induced development of SLE and EAE in mice start from three months of life, when the first signs of the onset of pathologies are observed [29,30,31,32].

It was shown that the spontaneous evolution of SLE and EAE occurs because of the specific reorganization of bone marrow hematopoietic stem cells (BM-HSCs). At the same time, the immunization of SLE-prone mice with DNA histone complexes and EAE-prone mice with MOG or DNA histone complexes causes a significant acceleration of the development of these AIPs [28,29,30,31,32]. In addition to specific profiles of changes in BM-HSC differentiation, a substantial rise in lymphocyte proliferation in various organs of mice occurs [28,29,30,31,32]. Moreover, these particular changes in the mouse immune system lead to the appearance of many different autoabzymes that split MOG, MBP, DNAs, RNAs, histones, and polysaccharides [28,29,30,31,32]. The detection of different abzymes is the statistically significant and earliest marker of the onset of many AIPs [24,25,26,28,29,30,31,32]. The enzymatic activities of Abs-abzymes are easily detected even at the onset of several AIPs (the pre-disease stage or the beginning of such pathologies) before the appearance of typical markers of specific AIPs (for review, see [24,25,26] and references therein). The titers of autoantibodies to specific autoantigens at the onset of many AIPs usually correspond to the typical range of indices that characterize conditionally healthy individuals. The appearance of plural abzymes indicates the start of autoimmune processes; a significant increase in the catalytic activities of Abs-abzymes is associated with the development of profound AIPs ([24,25,26,28,29,30,31,32] and references therein). However, several parallel mechanisms could indicate the development of different AIPs, usually leading to the breakdown of self-tolerance.

In addition to the mouse lines described above, which are predisposed to the development of EAE, several hybrid lines have been obtained using different EAE mice [38,39,40,41,42]. It was previously shown that in hybrid 2D2/Th mice, spontaneous development of EAE occurs in 46–51% of individuals on average six weeks after birth—earlier than in the Th and 2D2 lines [38,39]. Infiltration of CD4+ cells with the formation of foci of inflammation and demyelination in 2D2/Th mice was found mainly in the spinal cord and optic nerves.

Previously, as mentioned above, we analyzed the early stages of EAE development (3-month-old mice) associated with changes in the differentiation profiles stem cells in bone marrow and the production of DNA-, MBP-, and histone-hydrolyzing abzymes using the example of three lines of mice predisposed to the development of EAE-C57BL/6, Th, and 2D2 [29,30,31,32]. It was interesting to compare changes in the time course of EAE development in the above parameters in C57BL/6, Th, and 2D2 with those for hybrid 2D2/Th mice. For this purpose, an analysis of changes in the differentiation profile of stem cells in the bone marrow of 2D2/Th mice was carried out. IgGs were isolated from the blood of mice, followed by an analysis of their catalytic activities.

## 2. Results

### 2.1. Proteinuria Assay

Proteinuria (protein in urine > 3 mg/mL) may be an additional marker of very profound AIPs in mice [24,25,26,28,29,30,31,32]. Mice not predisposed to developing AIPs (BALB/c and (CBAxC57BL)F1) usually display a low level of 0.1–0.12 mg/mL proteinuria [28]. A completely different situation is observed in mice predisposed to the development of EAE [29,30,31,32]. The level of proteinuria is high at 3 months of age (6.6 and 7.6 mg/mL for 2D2 and Th mice, respectively). The proteinuria increases in the spontaneous development of EAE (Figure 1). Interestingly, in 3-month-old 2D2/Th mice, the concentration of proteins is 10.3 mg/mL and increases within 10 days to 13.3 mg/mL (Figure 1).

### 2.2. Hematopoietic Progenitor Colony Formation

As previously shown, the development of AIPs begins due to a change in the differentiation profile of BM-HSCs and an increase in the proliferation of lymphocytes in different organs [24,26,28,32]. The changes in the differentiation profile of stem cells in C57BL/6, 2D2, Th, and 2D2/Th mice were compared (Figure 2).

The evolution of EAE in C57BL/6 is very slow, and the data on different cells in these mice bone marrow may be roughly accepted as the norm for conditionally healthy mice that were used to create 2D2 and Th mice.

Interestingly, at 3 months of age in Th mice, the relative number of erythroid burst-forming unit early erythroid (BFU-E) colonies is 50 times greater than in 2D2 mice (*p* < 0.05; Figure 2A). Interestingly, hybrid 2D2/Th mice have 1.6-fold fewer (*p* < 0.05) of these cells than Th mice and a comparable number of these colonies compared to C57BL/6 mice. Moreover, Th mice show a very significant decrease in these cells during the spontaneous development of EAE, while 2D2 mice show a substantial increase in these cells. In hybrid mice, some of these cells grow for up to 10–15 days and then slowly decline. In general, hybrid mice exhibit, to some extent, the intermediate situation for BFU-E colonies found in 2D2 and Th mice. In C57BL/6 mice, the spontaneous progress in EAE will result in a very strong rise in BFU-E colonies (Figure 2A).

A different situation of changes in the relative number of colonies is observed in the case of burst colony-forming unit–late erythroid (CFU-E) cells (Figure 2B). In 3-month-old mice, the number of these colonies is less than in C57BL/6 mice (fold): Th (1.4), 2D2 (2.7), and hybrids 2D2/Th (3.5). During the progress of EAE, growth of these cells in C57BL/6 and a temporary increase in their number in Th mice were found. 2D2 and hybrid 2D2/Th mice show an almost smooth and parallel increase in the content of these cell colonies up to 60 days (Figure 2B).

The highest content of colony-forming unit granulocytic–macrophagic (CFU-GM) colonies was found in the bone marrow cerebrospinal fluid of hybrid 2D2/Th mice (Figure 2C). In 3-month-old mice, the content of these cells in 2D2 and Th mice is comparable and approximately 11-fold lower than in 2D2/Th mice (*p* < 0.05). At the same time, the number of these cells in C57BL/6 is only 1.5 times lower (*p* < 0.05) than in 2D2/Th mice. In C57BL/6 and 2D2, with the spontaneous development of EAE, a strong increase in the number of these cells is observed, and their number in Th mice remains almost unchanged over time. And only hybrid 2D2/Th mice show a slow decrease in the number of CFU-GM colonies.

At the beginning of the experiment, the largest number of colony-forming unit granulocytic–erythroid–megakaryocytic–macrophagic (CFU-GEMM) cells in 2D2/Th is approximately 8.7-fold higher than in 2D2 and 1.5 times higher than in Th and C57BL/6 mice (*p* < 0.05). Over 15 days of spontaneous progress of EAE in 2D2/Th mice, the number of colonies of such cells increases by 1.5 times (*p* < 0.05). However, the largest increase in these cells, 15.3-fold from time zero to 50 days, occurs in 2D2 mice. Interestingly, at time zero, the numbers of CFU-GEMM cells in Th and C57BL/6 mice are very close, but the development of EAE leads to their significant decrease.

Thus, the relative numbers of cells analyzed in the cerebrospinal fluid of the bone marrow of the four strains of mice at 3 months of age are very different. The nature of the change in their relative content during the development of EAE is also different. It should be noted that the content of CFU-GM and CFU-GEMM cells is much higher in hybrid 2D2/Th at three months of age compared to 2D2 and Th mice (Figure 2). One cannot exclude that the high content of these cells may be important in the accelerated development of EAE in hybrid mice compared with Th and 2D2 mice.

### 2.3. B Cells in Various Organs

As stated above, the development of EAE leads not only to a change in the differentiation profile of BM-HSCs but also to a change in the level of lymphocyte proliferation in different organs. Figure 3 shows a comparison of B lymphocyte content in different organs of Th, 2D2, and hybrid 2D2/Th mice.

The largest B cell concentration is in the blood of 3-month-old 2D2 mice, whose number increases with time (Figure 3A). The blood of Th and 2D2/Th mice is characterized by ~2-fold lower (<0.05) B cell levels, the content of which remarkably increases for Th but slowly decreases over time in 2D2/Th mice.

Bone marrow of 2D2 mice is also characterized by a high content of B cells (Figure 3B). Their amount is approximately 1.5 and 3.7 times lower than for Th and 2D2/Th mice, respectively. Considering the slight increase in these cells in 2D2 and 2D2/Th in 10–20 days, their decrease over time is observed. In the thymus, a higher content of B cells was also found in 2D2 mice, which is 3.6 times higher than in Th and 6.0 times higher than in 2D2/Th mice (*p* < 0.05). If the content of these lymphocytes increases in 2D2 and Th, then it changes very little in 2D2/Th mice.

The number of B cells in the spleens of Th and 2D2 mice is somewhat comparable and changes slightly over time (Figure 3B). In three-month-old 2D2/Th mice, the content of B lymphocytes is approximately 1.8–2.2 times lower than in Th and 2D2 mice, and by 20 days of EAE development, it decreases by approximately two times.

At time zero of the experiment, the relative contents of B cells in the lymph nodes of 2D2, Th, and 2D2/Th mice differ slightly. However, with the development of EAE, their content increases strongly in 2D2 but changes slightly in Th mice. However, in the case of 2D2/Th mice, their number decreases by approximately 4.6 times by 30 days of EAE development.

In general, the highest content of B cells in all organs was found in 2D2 and the lowest in 2D2/Th hybrid mice. The cell content in Th mice is closer to the average of these values for 2D2 for 2D2/Th mice (Figure 3).

In addition, only 2D2/Th mice tended to decrease the B cells’ content during the development of EAE.

### 2.4. T Cells in Organs

Figure 4 compares the content of T lymphocytes in different organs of three lines of EAE-predisposed mice.

In contrast to B cells, the maximum level of T lymphocytes was found in 2D2/Th hybrid mice in the blood (A) and thymus (C): these dependencies in all cases have a bell-shaped character with maximums at approximately 10–30 days. The content of T lymphocytes in the blood of 2D2 and Th mice, respectively, is approximately 1.4 and 4.0 times lower (*p* < 0.05) than that of 2D2 mice. The same situation occurs in the case of the thymus, in which content of T cells is 1.3 and 4.5 times lower.

In the lymph nodes, the concentrations of T cells in three lines of three-month-old mice are comparable (E). At the same time, in 2D2/Th and 2D2 mice, these cells sharply increase by approximately 20 days and then decrease. In the case of Th mice, their relative amount changes relatively little over time (E).

Somewhat unexpectedly, the content of T lymphocytes in the blood and thymus of 2D2/Th is much higher than that of 2D2 and Th used to obtain these hybrids. The content of T cells in the bone marrow and spleen of 2D2/Th mice is the lowest compared to 2D2 and Th mice; their content tends to increase during the development of EAE. With different initial concentrations of T cells in the bone marrow of 2D2, Th, and 2D2/Th at three months of age, the development of EAE leads to a decrease in these indexes.

Thus, the relative contents of T and B lymphocytes in different organs of 2D2, Th, and 2D2/Th are very different. In general, there is a low content of B cells in various organs of 2D2/Th mice compared to that for 2D2 and Th mice. At the same time, the content of T lymphocytes in the blood and thymus of 2D2/Th mice is significantly higher than in 2D2 and Th mice. Nevertheless, in the spleen, the relative concentration of T cells in 2D2/Th is lower than that of the 2D2 and Th mice, and their change patterns are different.

### 2.5. The Relative Content of Different Abs

The relative content of different Abs was measured using standard ELISA and blood plasma. The blood of healthy mammals contains auto-Abs against DNA and various proteins in low concentrations [30,31,32]. We estimated the relative concentration of Abs against DNA, MBP, MOG, and histones (Figure 5).

The relative concentration of antibodies against DNA (at 3 months of age) in sera of 2D2/Th mice was 7- and 12-fold higher (*p* < 0.05) than in Th and 2D2 mice, respectively (Figure 5A). A sharp increase in the concentration of anti-DNA antibodies in 2D2/Th mice occurred in two stages at 15 and 60 days of spontaneous development of EAE. At the same time, the concentrations of these antibodies in 2D2 and Th mice changed slightly during the development of EAE. For C57BL/6, a more substantial change in the concentration of anti-DNA antibodies over time was observed (Figure 5A).

The fact that 2D2/Th demonstrates an increased concentration of antibodies against DNA already in 3-month-old mice and the development of EAE within 90 days may play an important role in the accelerated development of this pathology in 2D2/Th mice. As has been shown previously, some antibodies against DNA have DNase activity [24,25,26,28,29,30,31,32]. DNA-hydrolyzing abzymes penetrate the membranes of cells and nuclei and hydrolyze chromatin DNA, stimulating cell apoptosis [43,44]. This leads to an increase in the concentration of DNA complexes with histones in the blood. However, DNA complexes with histones are the main antigens in the formation of anti-DNA antibodies [45]. An increase in the concentration of such antibodies leads to increased cell apoptosis and the flare-up of autoimmune processes [24,25,26].

The concentration of anti-MBP antibodies in the blood of 2D2 and Th mice is significantly lower than in the case of C57BL/6 mice (Figure 5B). Moreover, the relative content of anti-MBP antibodies in 2D2/Th mice is approximately comparable to the average value of these parameters for 2D2 and Th mice.

Particular attention should be paid to higher concentrations of antibodies in 3-month-old 2D2/Th mice against MOG compared to 2D2 (2.1-fold) and Th (18.6-fold) and C57BL/6 (16.0-fold) mice (Figure 5C). In addition, a high level of antibodies against MOG in 2D2/Th persists throughout the development of EAE. It is known that MOG is widely used as an inducer activating the development of EAE in the case of several strains of mice predisposed to EAE [33,34,35,36,37,38]. However, the blood of mice also contains internal MOG. Thus, due to the increased concentration of MOG in the blood of 2D2/Th mice, the development of EAE may be accelerated.

At time zero of the experiment, the relative concentrations of antibodies against 5 histones in 2D2, Th, and 2D2/Th are somewhat comparable and significantly lower than in C57BL/6 mice (Figure 5D). The concentration of Abs against five histones increases very strongly in 2D2/Th mice during the development of EAE, while it changes very weakly in 2D2 and Th mice.

This indicates that the development of EAE leads to an increase in the concentration of histones in the blood of 2D2/Th mice. An important factor in the acceleration and deepening of EAE in 2D2/Th mice may be due not only to antibodies against histones but also to the histones themselves. Free extracellular histones in the blood usually act as damage factors [46,47]. Mouse immunization using exogenous histones determines several toxic effects due to inflammatory reactions and activation of Toll-like receptors leading to lethal endotoxemia, trauma, ischemia–reperfusion injury, pancreatitis, peritonitis, stroke, coagulation, thrombosis, and progression of autoimmune reactions [46,47].

### 2.6. Catalytic Antibodies

As noted above, abzymes play an important role in the development of EAE as they appear in the blood after a change in the differentiation profile of BM-HSCs [46,47]. To analyze the enzymatic activities of antibodies, they were isolated using a previously described method, which ensured the absence of classical enzymes in the resulting preparations.

Similar to [28,29,30,31,32], using strict criteria, antibodies from 2D2/Th hybrid mice were shown to be electrophoretically homogeneous and free of enzyme contaminants (Figure 6).

After SDS-PAGE, IgGs of 2D2/Th were analyzed similarly to [28,29,30,31,32], and the positions of IgGs hydrolyzing DNA, MOG, MBP, and histones correspond only to small fragments of gel containing intact 150 kDa IgGs. These data indicated the absence of impurities of canonical DNases and proteases in antibody preparations.

Such antibodies were used to analyze changes in their activities during the development of EAE in 2D2/Th mice (Figure 7).

At time zero of the experiment, the DNase activity of IgG (Figure 7A) was comparable for 2D2 and hybrid 2D2/Th and approximately 28 times higher than in Th and C57BL/6 mice (*p* < 0.05). The development of EAE in Th and C57BL/6 mice led to a slow and weak increase in the DNA-hydrolyzing activity of IgGs in these mice. By 14 days, DNase activity in 2D2 mice IgGs increased ~2.2 times, while in 2D2/Th, there was a slight change. However, by 30–60 days in 2D2/Th mice, DNase activity of antibodies increased 11 times (*p* < 0.05). This increase in DNase activity at 50–60 days (Figure 7A) correlates with a strong increase in the concentration of anti-DNA antibodies by 50–60 days of EAE development in 2D2/Th (Figure 5A).

As mentioned above, abzymes with DNase activity play a very negative role in fueling autoimmune processes. Taking this into account, the high DNase activity of abzymes in 2D2/Th mice can stimulate accelerated and deeper processes in the development of EAE pathology.

Figure 7B shows data on the hydrolysis of MBP. As in the case of DNase activity at three months of age, IgG antibodies of 2D2 and 2D2/Th demonstrate comparable activity in the hydrolysis of MBP. However, there is then a marked increase in the protease activity of 2D2 mice IgG antibodies but a sharp decrease in this activity for IgGs of hybrid 2D2/Th mice. At time zero of the experiment, MBP-hydrolyzing activity of antibodies in C57BL/6 and Th mice is, respectively, 2.1 and 8.5 times lower (*p* < 0.05) than in 2D2/Th, and then there is a significant increase in antibody activity in both strains of mice.

Data on hydrolysis of MOG with antibodies corresponding to four different strains of EAF mice are shown in Figure 7C. The relative activity of MOG-hydrolyzing IgGs is comparable for 3-month-old C57BL/6 and 2D2/Th mice and is approximately 2.0- and 6.9-fold higher (*p* < 0.05) than for 2D2 and Th mice, respectively (Figure 7C). With the development of EAE in C57BL/6 and Th mice, there is a strong increase in the protease activity of abzymes. At the same time, for 2D2, in the period 0–14 days, there is a significant increase in the activity of MOG hydrolysis. The pattern of changes in the efficiency of MOG hydrolysis by antibodies of hybrid 2D2/Th mice has a specific character. From time zero to 20 days, there is a strong decrease in the activity of abzymes, and then it increases at 30–40 days and again there is a decrease.

As stated above, immunization of mice with MOG greatly accelerates the development of EAE in all strains of mice predisposed to this pathology. As can be seen from Figure 5, the concentration of intrinsic MOG in the blood of 2D2/Th is much higher than that of the other three strains of mice. This may result in increased levels of anti-MOG antibodies with and without catalytic activity. Such MOG and anti-MOG antibodies may be important for the accelerated and more profound development of EAE.

### 2.7. Hydrolysis of Histones

As mentioned above, free extracellular histones [46,47] and anti-histone antibodies hydrolyzing histones and MBP [24,25,26] are very damaging factors. We compared the activity of IgGs against five histones in histones hydrolysis for four strains of EAE-prone mice (Figure 8).

Histone H1 hydrolysis at time zero was highest for IgGs of 2D2/Th hybrid mice compared with Th > C57BL/6 > 2D2 (Figure 8A). The activity of IgGs of 2D2 mice was 18 times lower than that of 2D2/Th mice (*p* < 0.05) and changed very little over time. The protease activity of antibodies of Th mice decreased by approximately 7 times but increased 3 times for C57BL/6 mice by day 70. In the case of hybrid 2D2/Th mice, a complex dependence of changes in the activity of antibodies in the hydrolysis of H1 histone was observed. By the 10th day of the experiment, it decreased by ~10 times, then there was a sharp increase compared with the activity of 10 days by 17 times, and its decrease began again.

In 3-month-old mice, antibody activity in hydrolysis of H2A histone in the case of Th and C57BL/6 mice was comparable and 1.4–1.6 times lower than in 2D2 mice (Figure 8B). At the same time, in 2D2 and C57BL/6, an increase in their activity was observed by 10–20 days, and in Th mice, there was a noticeable decrease in the activity. The lowest activity, 2.1-fold lower than that of 2D2, was found in hybrid 2D2/Th mice, which strongly decreased over time as in Th mice.

In the hydrolysis of histone H2B, the highest activity was found for 2D2/Th, which at time zero was higher than in mice of the three other strains (fold): 2D2 (9.2), Th (6.5), and C57BL/6 (3.1) (*p* < 0.05). By 20 days, a sharp increase in antibody activity was detected in the splitting of H2A for 2D2 and C57BL/6, while Th mice demonstrated a slight change in this activity over time. For 2D2/Th mice, the dependence of changes in the activity of anti-H2B abzymes was complex: first, there was a very sharp decrease in H2B hydrolysis by 10 days, and then there was a significant increase.

Only for Abs from C57BL/6 mice, which exhibit the lowest activity in the hydrolysis of H3 histone, is a slow increase in their activity over time observed. The dependencies are complex for antibodies of the other three strains of mice. For abzymes of 2D2 and Th mice, their activity first increases by 10 days, and then a strong decrease occurs. In 2D2/Th hybrid 3-month-old mice, antibody activity in histone H3 hydrolysis is higher than in the other three strains of mice. Then, after a decrease of approximately 10 days, it increases slowly.

The dependencies of histone H4 hydrolysis by IgGs of four lines of EAE mice are very complex. In the case of two lines (2D2 and Th mice), from time zero to 10–20 days of the experiment, a sharp increase in abzyme activity is observed, and then a sharp decline. For C57BL/6 and 2D2/Th hybrid mice, in the period 0–14 days, there is a sharp decrease in antibody activity, and then it sharply increases by 20 days.

### 2.8. Paralyzed 2D2/Th Mice

According to the description of the characteristics of 2D2/Th mice [38,39,40,41,42], spontaneous development of EAE occurs in 46–51% of individuals on average six weeks after birth. For comparison with 2D2 and Th mice, which were used to generate hybrid mice, 3-month-old mice were used in this work. The peculiarity of 2D2/Th mice was that approximately 4–5% of them became paralyzed with loss of the ability to move at about 3 months of age. Taking this into account, we analyzed some parameters in paralyzed mice (Figure 9).

Figure 9A compares the relative activity of IgGs in DNA hydrolysis for antibody preparations of paralyzed 2D2/Th hybrids, as well as non-paralyzed 2D2/Th, 2D2, and Th mice. It can be seen that the DNase activity of antibodies from the blood of paralyzed mice is ~5.8 times higher (*p* < 0.05) than that of non-paralyzed 2D2/Th mice. Interestingly, the DNase activity of IgGs from paralyzed 2D2/Th mice is approximately 4.7 and 172.0 times higher than in control 2D2 and Th mice (*p* < 0.05).

A similar situation was observed for MOG-hydrolyzing activity, which in the case of paralyzed mice is 4.5-fold higher than for non-paralyzed 2D2/Th mice (Figure 9B). In control 2D2 and Th mice, protease activity is, significantly, 5.9 and 4.2 times (*p* < 0.05) lower than in paralyzed 2D2/Th mice.

At the same time, there is only a tendency for the activity of antibodies to increase in the hydrolysis of MBP in paralyzed 2D2/Th compared to non-paralyzed 2D2/Th and 2D2 mice, but the difference is not significant (*p* > 0.05). There is only a strong increase in activity compared to Th mice. Most likely, it is the activity of antibodies in the hydrolysis of DNA and MOG, but not MBP, which may be significant in achieving paralysis in hybrid mice.

## 3. Discussion

Various autoimmune and neurodegenerative diseases were demonstrated to begin after changes in the differentiation profile of BM-HSCs, leading to the production of harmful abzymes that hydrolyze various components of cells and tissues [23,24,25,26,28,29,30,31,32]. It was exciting to analyze the changes in various specific parameters characterizing the spontaneous development of EAE in hybrid 2D2/Th mice compared with the 2D2 and Th mice used for their generation.

The level of proteinuria, which may be an additional marker of very profound AIPs in mice [24,25,26,28,29,30,31,32] was high at 3 months of age in all four lines of mice and it increases during the development of EAE (Figure 1).

Figure 2 demonstrates that the relative contents of hematopoietic bone marrow progenitors in the bone marrow of four mouse strains at three months of age are very different. Moreover, in 2D2/Th hybrids, only in the case of BFU-E cells, their content is approximately comparable with the average value of this parameter for 2D2 and Th mice. At the same time, the number of other types of CFU-E, CFU-GM, and CFU-GEMM cells in 2D2/Th mice may be more or less than in 2D2 and Th mice. In connection with this, it should be noted that the content of CFU-GM cells in 2D2/Th hybrids is significantly higher at three months of age and changes slightly during the development of EAE (Figure 2C). However, the bone marrow number of CFU-E in 2D2/Th mice is the lowest and changes little over time (Figure 2B). The relative amounts of four types of bone marrow cells change during the development of EAE in all lines of mice. However, the changes in the relative numbers of BFU-E, CFU-E, CFU-GM, and CFU-GEMM cells in 2D2/Th are not comparable with the average value of these parameters for 2D2 and Th mice. For example, a decrease in CFU-GM cells over time occurs in 2D2/Th and Th, while in 2D2, there is growth over time. In other words, each line of EAE mice demonstrates the specific content of all four types of hematopoietic cells at three months of age and the particular nature of their changes during the development of EAE. Despite the differences in changes in the differentiation profiles of BM-HSCs in different lines of mice, all of them develop EAE. At the same time, differences in changes in the differentiation of bone marrow stem cells can lead to significant differences in the lymphocytes producing abzymes that are harmful to the body to varying degrees.

Particular attention should be paid to the very high concentration of antibodies against DNA (Figure 5A) and MOG (Figure 5C) in 2D2/Th compared to the three other mouse strains. In addition, a particularly strong increase during the development of EAE is observed in DNA-hydrolyzing IgGs (Figure 8A). As previously shown, anti-DNA antibodies penetrate through the cellular and nuclear membranes, causing cell apoptosis [43,44], which leads to an increase in the concentration of blood DNA complexes with histones and acceleration of the development of AIPs [24,25,26]. The fact that 2D2/Th demonstrates an increased concentration of antibodies against DNA already in 3-month-old mice and an increase in the DNAse activity within 90 days may play an important role in the accelerated development of this pathology in 2D2/Th mice.

In addition, MOG is an accelerator of the development of pathology in all EAE-prone mouse strains [33,34,35,36,37,38,39,40,41,42]. The blood of 4–5% of 2D2/Th mice, which become paralyzed by three months of age, contains antibodies 5.8-fold more active in DNA hydrolysis than non-paralyzed mice (Figure 9). Thus, the increased content of anti-MOG abzymes and antibodies with DNase activity in the blood of 2D2/Th may play an important role in developing pathology in these mice.

Interestingly, histones and abzymes against them can play an additional important role in the pathogenesis of EAE in all strains of mice, especially in 2D2/Th hybrids. As shown previously [46,47], the introduction of histones into the blood of mice leads to the development of various pathologies, including AIPs. Moreover, the complexes of histones with DNA are the main antigens that determine the production of antibodies against DNA and histones [45]. As can be seen from Figure 5D, only in 2D2/Th mice is there a significant increase in the concentration of antibodies against histones during the development of EAE, and by 60 days, it increases 11 times compared to time zero. During the development of EAE, there are special time intervals when there is a sharp increase in the activity of antibodies of 2D2/Th mice in the hydrolysis of several of the five histones (Figure 8).

It is believed that AIP development may be stimulated by foreign antigens of different bacteria or viruses [48,49,50,51]. Molecular mimicry due to homology between human and viral molecules, including Epstein–Barr, measles, hepatitis B, herpes simplex, influenza, and papilloma viruses, may participate in the autoimmune pathogenesis of MS [48,49,50,51]. Different antigens of some viruses or bacteria can penetrate through the blood–brain barrier and stimulate specific changes in the BM-HSCs, leading to the production of Abs against such antigens. During long periods of illness, there may be specific switching of the immune system to the synthesis of B lymphocytes that produce antibodies against their own antigens [48,49,50,51]. This statement is supported by data from studies [52,53] where it was shown that the bone marrow coelomic fluid of some MS patients contains oligoclonal antibodies against viral proteins. However, humans and animals have internal mechanisms that can stimulate the development of autoimmune reactions.

As shown in a number of works [24,25,26], the destruction-hydrolysis of MBP in the membranes of nerve tissues plays a very important role in the development of MS in humans and EAE in experimental animals. However, hydrolysis of MBP can occur not only under the action of abzymes against MBP but also against five histones. A number of studies have shown that due to the high level of homology between H1, H2A, H2B, H3, and H4 histones and MBP, abzymes against each of these histones hydrolyze any of the five histones and MBP, and vice versa, antibodies against MBP hydrolyze MBP and all five histones [24,25,26,28,29,30,31,32]. One of the mechanisms for removing lymphocytes that produce antibodies harmful to the body is cell apoptosis [24,25,26]. The blood constantly contains antibodies against histones, including abzymes, which hydrolyze not only histones but also MBP of the membranes of nerve tissues. Thus, people have an additional internal mechanism for the possibility of MS development.

## 4. Material and Methods

### 4.1. Reagents

Protein G-Sepharose, Superdex 200 HR 10/30 column, all proteins, and other different compounds and reagents were from GE Healthcare (New York, NY, USA) and Sigma-Aldrich (Munich, Germany). The human MBP of 18.5–14.5 kDa was from RCMDT (Moscow, Russia), while MOG_35–55_ was from EZBiolab (Heidelberg, Germany). All preparations were free from possible contaminants.

### 4.2. Choosing a Model for Analysis

The evolution of EAE in Th and 2D2 mice arises spontaneously and proceeds relatively slowly [24,25,26]. Treatment of C57BL/6, Th, and 2D2 mice with DNA complexes with histones or MOG greatly accelerates the development of EAE [24,25,26,28,29,30,31,32]. Different clinical and neurological indicators usually appear at relatively late stages of the development of EAE pathology in mice [24,25,26,28,29,30,31,32]. At the same time, a change in the differentiation profile of bone marrow stem cells and the production of abzymes that hydrolyze different autoantigens occurs in the early stages during the spontaneous development of EAE in mice [24,25,26,28,29,30,31,32]. After EAE mice treatment with MOG or DNA–histone complexes, several stages in the process of EAE were revealed: the onset, beginning at 7–8 days, the acute phase at 18–20 days, and the following remission stage at 25–30 days [24,25,26,28,29,30,31,32]. The change in cell differentiation profile results in the production of adverse lymphocytes synthesizing abzymes, destroying DNAs, RNAs, MBP, MOG, and histones. The parameters characterizing these specific changes in C57BL/6, Th, and 2D2 mice were investigated earlier [28,29,30,31,32]. In this article, we analyzed several of the parameters mentioned above characterizing spontaneous development in hybrid 2D2/Th mice and compared them with those of C57BL/6, Th, and 2D2 mice.

Protein in urine of mice was measured using a standard Bradford, Lowry method as in [24,25,26,28,29,30,31,32].

### 4.3. Experimental Animals

The C57BL/6, inbred 2D2 TCR (TCR^MOG^), and Th 3-month-old mice were described earlier including genotypic, phenotypic, and other differences in [29,30,31,32]. All these mice were obtained from Prof. Sven Meuth (Münster; Westfälische Wilhelms-Universität, Department of Neurology, (Germany, Munster) and Prof. Thomas Budde (Münster; Westfälische Wilhelms-Universität, Germany, Munster). These mice were kept in a special mouse breeding facility free of pathogens.

Hybrid 2D2/Th mice were obtained by crossing of Th and 2D2 mice using the standard method. The characteristics of these hybrid mice were consistent with those described previously [38,39].

A group of 70 three-month-old males was used, which were divided into subgroups of 7 mice. Each group of mice was decapitated at different times after the start of the experiment from 0–90 days (Figure 2, Figure 3, Figure 4 and Figure 5, Figure 7, and Figure 8).

All experiments complied with the Bioethical Committee of the Institute of Chemical Biology and Fundamental Medicine (Number Protocol 21-4 from 15 August 2020) under all principles of the Directive of the European Communities Council (86/609/CEE) for working with animals. The Bioethical Committee of the institute supported our study.

### 4.4. Bone Marrow Progenitor Cell Analysis in Culture

Preparations of the bone marrow of hybrid 2D2/Th mice were obtained from mouse femurs similar to 2D2 and Th mice [29,30,31,32]. The growth of bone marrow cells forming colonies was estimated as in [29,30,31,32]. Cells (2 × 10^4^) from every mouse were grown in standard conditions using four dishes and a special methylcellulose-based M3434 medium (CanadaStemCell Technologies; Vancouver, BC, Canada). This medium was supplemented with IL-6, erythropoietin, and IL-3. After fourteen days of incubation at 37 °C in a humidified incubator (5% CO_2_), the relative content of CFU-GEMM, CFU-GM, BFU-E, and CFU-E cell colonies on 4 dishes for every mouse was calculated as in [29,30,31,32]. The content of B and T lymphocytes in various organs of 2D2/Th mice was determined by flow cytometry as in [29,30,31,32] (see Appendix A).

### 4.5. ELISA of Anti-Antigen Antibodies

The relative concentrations of Abs against MBP, MOG, five histones, and DNA in plasma samples were estimated similarly to [29,30,31,32]. After treating immobilized plasma components with specific anti-mouse Abs conjugated with horseradish peroxidase, the mixtures obtained were incubated with H_2_O_2_ and tetramethylbenzidine. The optical density of the mixtures (A_450_) after the addition of H_2_SO_4_ was measured by the Uniskan II plate reader (MTX Lab System, Vienna, VA, USA) [29,30,31,32]. The A_450_ values were estimated taking into account differences in A_450_ of experimental and control mixtures containing no MBP, MOG, histones, or DNA.

### 4.6. IgG Purification

Electrophoretically homogeneous IgGs from hybrid 2D2/Th mice were purified using plasma protein chromatography on Protein G-Sepharose and gel filtration on Superdex 200 in harsh acidic (pH 2.6) conditions [29,30,31,32]. IgG preparations were protected from contamination by filtration using 0.1 μm Millex membranes. SDS-PAGE of IgG preparations was performed using 5–15% gradient gels and visualized by silver or Coomassie blue staining [29,30,31,32].

### 4.7. DNA-Hydrolyzing Activity Assay

Activity of IgGs in the hydrolysis of DNA was estimated similar to [29,30,31,32]. The reaction mixtures (15–25 μL) consisted of Tris-HCl buffer (20 mM, pH 7.5), 20.0 μg/mL supercoiled (sc) DNA (pBluescript), MgCl_2_ (5.0 mM), 1.0 mM EDTA, 20.0 mM NaCl, and 0.01–0.1 mg/mL of IgGs. After incubation of solutions for 3.0–25 h at 37 °C, products of supercoiled DNA degradation were analyzed by electrophoresis using 0.8% agarose gels. The gel coloration was performed using ethidium bromide, and their analysis was carried out by Gel-Pro Analyzer v9.11 (Media Cybernetics, Rockville, MD, USA). The relative DNase activities (RAs) were found by comparing intact supercoiled DNA and its degraded-relaxed form. All initial rates of supercoiled DNA splitting were calculated using linear parts of the curves of scDNA hydrolysis (30–40%) and linear fragments of the rate dependencies on IgG concentrations. A complete hydrolysis of scDNA to its relaxed forms was taken for 100% of DNA-hydrolyzing activity. Using pseudo-first-order reaction conditions and only linear parts of the dependencies allows for recalculating the relative activities in the same conditions. The relative activities (RA, % of the DNA hydrolysis) finally correspond to the same standard time and concentration of IgGs as in [29,30,31,32].

### 4.8. Protease Activity Assay

For protease activity investigation, reaction mixtures (10–60 μL) consisted of Tris-HCl (20 mM, pH 7.5), 1.0 mg/mL histones, MOG, or MBP and IgGs (0.001–0.2 mg/mL) similar to [29,30,31,32]. After mixture incubation for 5.0–25 h (37 °C), products of histones, MOG, and MBP hydrolysis were detected after SDS-PAGE using a 5–15% gradient or 12% gels and staining with Coomassie R250. After scanning the gels, the hydrolysis products were quantified by GelPro v3.1 software. The relative protease activities of all IgG samples were calculated from a decrease in initial proteins (%). The hydrolysis of histones, MOG, or MBP incubated without IgG preparations was taken into account. All protein splitting rates were estimated using pseudo-first-order reaction conditions: linear parts of time dependencies and concentrations of IgG preparations (20–40% of the hydrolysis).

### 4.9. SDS-PAGE Analysis of Catalytic Activities

The SDS-PAGE assay of DNase and protein-degrading activities of IgG preparations was performed similarly to [29,30,31,32]. IgG preparations were first incubated at 25 °C for 32 min using 50 mM Tris-HCl buffer (pH 7.5) containing 1% SDS and 10% glycerol (non-reducing conditions). To restore the enzymatic activities of IgGs after SDS-PAGE, the gels were incubated for 70 min at 20 °C with 4.0 M urea and washed 7 times (10–15 min) with water to remove SDS. Cross-sections of longitudinal gel slices (2–3 mm) were cut and then incubated with Tris-HCl (55 μL; 50 mM, pH 7.5, containing 40 mM NaCl) for 4–7 days at 4 °C to allow refolding of molecules of IgG and elution from the gel. The solutions were separated from gel particles by centrifugation and used as described above to analyze DNA- and protein-hydrolyzing activities. Parallel longitudinal control strips of gels were used to detect possible positions of IgGs or other proteins on the gel using silver or Coomassie R250 staining.

All reported values are presented as the mean and SD of at least 2–3 independent experiments. Some sets of samples did not match the Gaussian distribution. Therefore, the Mann–Whitney U test was used to estimate the differences between various analyzed parameters; *p* < 0.05 was considered statistically significant.

## 5. Conclusions

This article is the first to investigate the changes in the differentiation profiles of bone marrow hematopoietic stem cells (BM-HSCs) and the level of B and T lymphocytes in different organs of hybrid 2D2/Th compared to 2D2 and Th mice. The faster and deeper spontaneous development of EAE in 2D2/Th is associated with another profile of BM-HSC differentiation, and with significant differences in the proliferation of B and T lymphocytes in the blood, bone marrow, thymus, spleen, and lymph nodes compared with 2D2 and Th mice. These changes lead to the production of Abs-abzymes against self-antigens: myelin oligodendrocyte glycoprotein (MOG), DNA, myelin basic protein (MBP), and five histones (H1–H4). The patterns of changes are significantly or radically different for 2D2/Th compared with control mice. Several factors may play an important role in the acceleration and deepening of EAE in 2D2/Th mice. It is known that mice treated with MOG show accelerated EAE development. In the initial stage of EAE development, the concentration of anti-MOG Abs in 2D2/Th is significantly higher than in Th (29.1-fold) and 2D2 (11.7-fold) mice. As shown earlier, DNA-hydrolyzing Abs-abzymes penetrate cellular and nuclear membranes and stimulate cell apoptosis, which leads to autoimmune processes. In the initial stage of EAE development, the concentration of anti-DNA Abs in 2D2/Th hybrids is higher than in Th (4.6-fold) and 2D2 (25.7-fold) mice. Only in 2D2/Th mice was there a very strong 10.6-fold increase in the DNase activity of IgGs during the development of EAE. Free histones in the blood are known to be cytotoxic, stimulating the development of autoimmune diseases. Only in 2D2/Th mice, during different stages of EAE development, is a sharp increase in the anti-histone activity in the hydrolysis of some histones observed. Thus, the development of EAE in hybrid 2D2/Th mice occurs similarly to that for 2D2 and Th but is characterized by significant differences in cellular and immunological parameters.

## Figures and Tables

**Figure 1 ijms-25-09900-f001:**
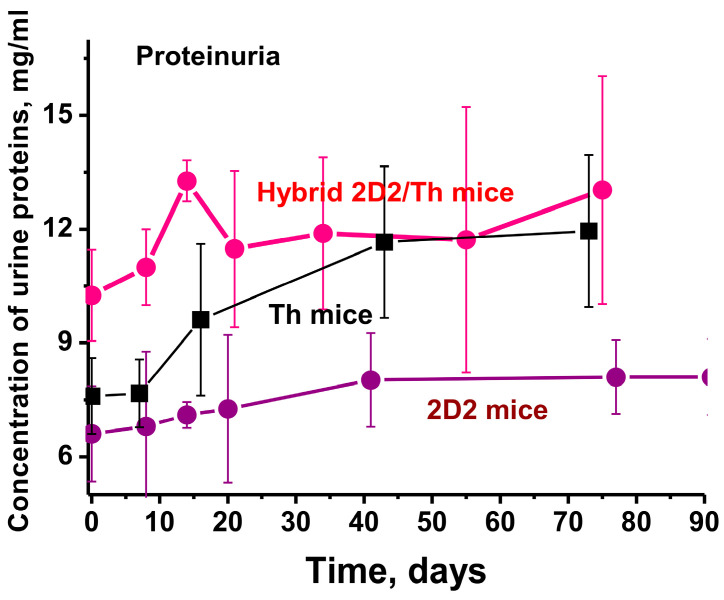
Comparison of proteinuria levels in 3-month-old C57BL/6, 2D2, Th, and hybrid 2D2/Th mice. For comparison, data for 2D2 and Th mice are taken from the works [29,30,31,32]. The difference between the data of all curves is characterized by *p* < 0.05.

**Figure 2 ijms-25-09900-f002:**
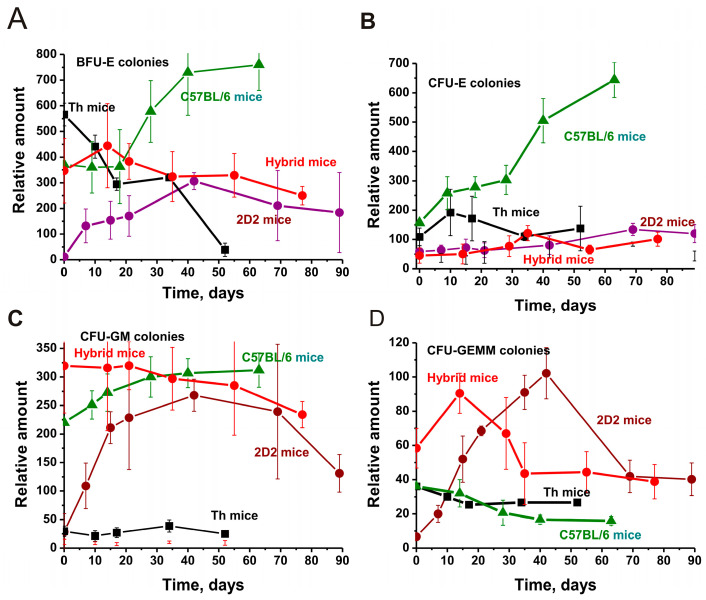
Changes in BFU-E (**A**), CFU-E (**B**), CFU-GM (**C**), and CFU-GEMM (**D**) cell colony units during development of EAE in C57BL/6, Th, 2D2, and hybrid 2D2/Th mice. The relative number of all colonies was calculated for 15,000 bone marrow cells. For comparison, data for C57BL/6, 2D2, and Th mice are taken from the works [29,30,31,32]. The difference between the data of all curves is characterized by *p* < 0.05.

**Figure 3 ijms-25-09900-f003:**
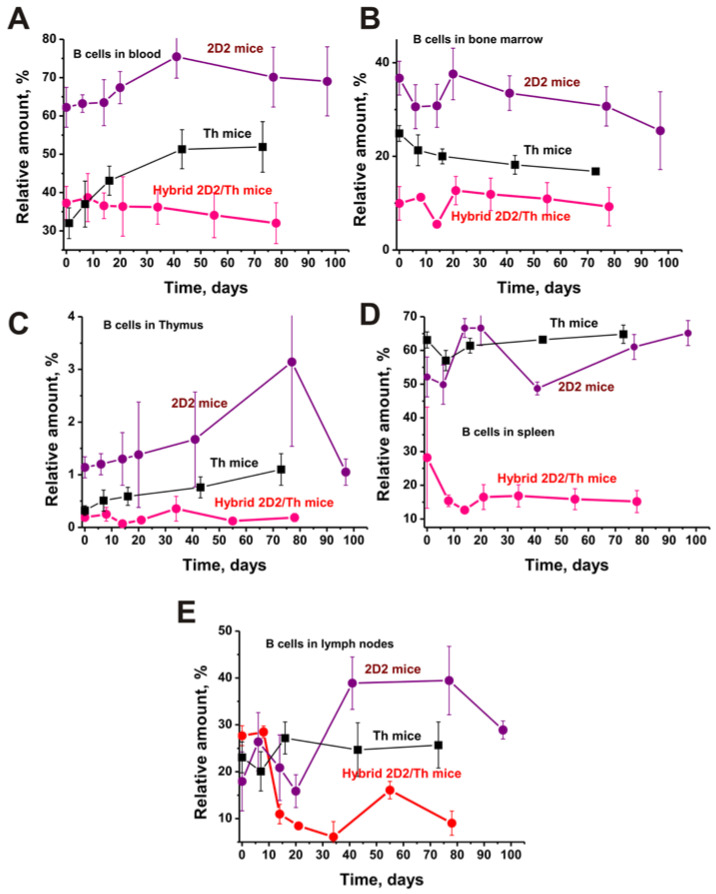
Changes in the content of B cells in different organs of Th, 2D2, and 2D2/Th mice: blood (**A**), bone marrow (**B**), thymus (**C**), spleen (**D**), and lymph nodes (**E**). Data for 2D2 and Th mice are taken from the works [29,30,31,32] for comparison. The difference between the data of all curves is characterized by *p* < 0.05.

**Figure 4 ijms-25-09900-f004:**
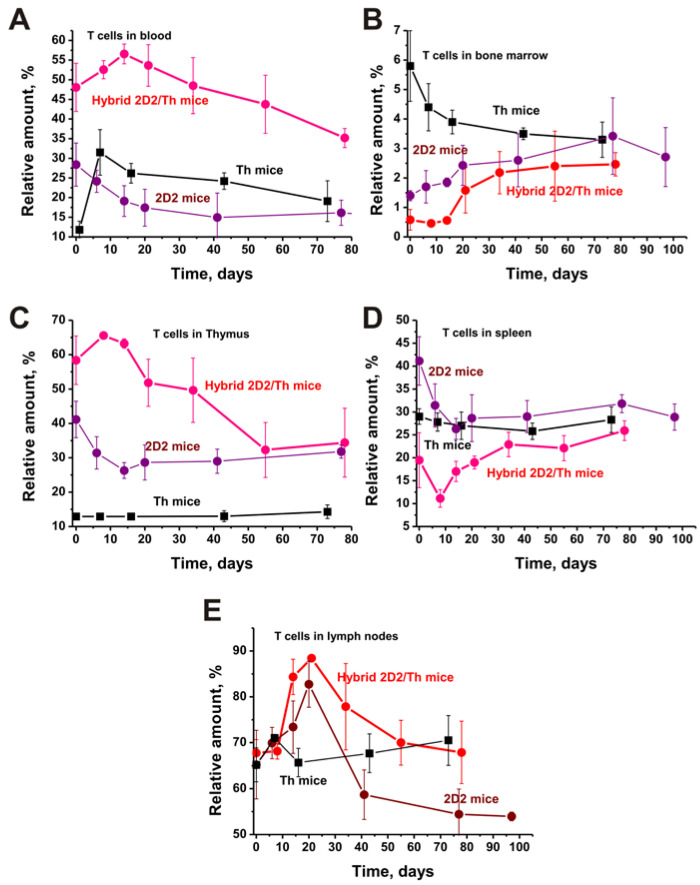
Changes in the T cell content in different organs of Th, 2D2, and 2D2/Th mice: blood (**A**), bone marrow (**B**), thymus (**C**), spleen (**D**), and lymph nodes (**E**). For comparison, data for 2D2 and Th mice are taken from the works [29,30,31,32]. The difference between the data of all curves is characterized by *p* < 0.05.

**Figure 5 ijms-25-09900-f005:**
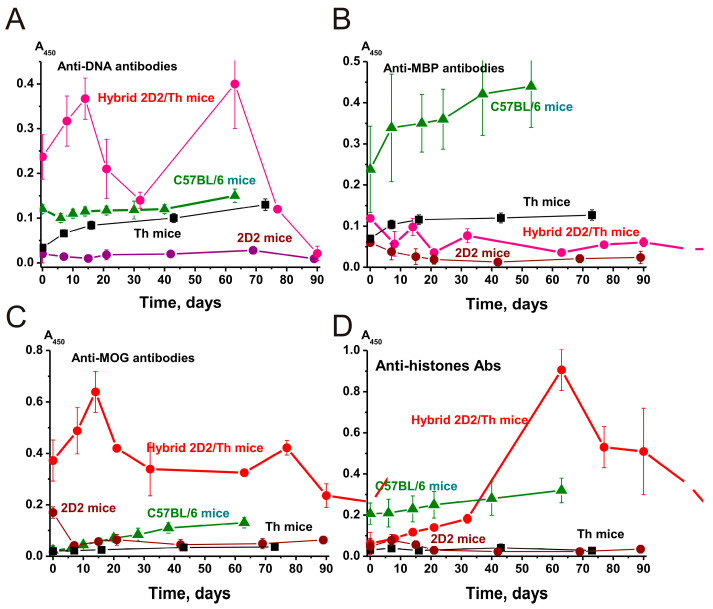
The changes in the concentration of antibodies against DNA (**A**), MBP (**B**), MOG (**C**), and five histones (**D**) during development of EAE in C57BL/6, Th, 2D2, and hybrid 2D2/Th mice. Dependencies corresponding to different lines of mice are marked in the panels. For comparison, data for C57BL/6, 2D2, and Th mice are taken from the works [29,30,31,32]. The difference between the data of all curves is characterized by *p* < 0.05.

**Figure 6 ijms-25-09900-f006:**
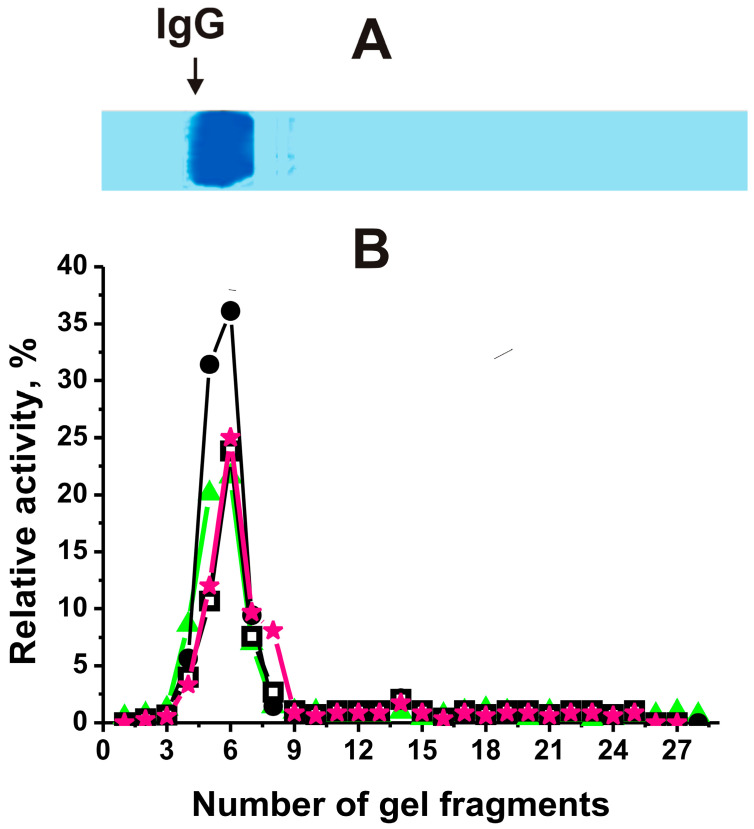
The IgG_mix_ (mixture of 21 IgG preparations; 17 μg) homogeneity analysis by SDS-PAGE with Coomassie staining (**A**). Panel (**A**) demonstrates the position of IgG_mix_. The relative activities in the hydrolysis of DNA (•), MOG (□), MBP (★), and histones (∆) were determined using eluates of many gel fragments (2–3 mm) (**B**). After substrate incubation for 24 h with the eluates from gel, complete hydrolysis of four antigens was taken for 100% (**B**). The errors of the activities from 2 independent experiments did not exceed 7–10%.

**Figure 7 ijms-25-09900-f007:**
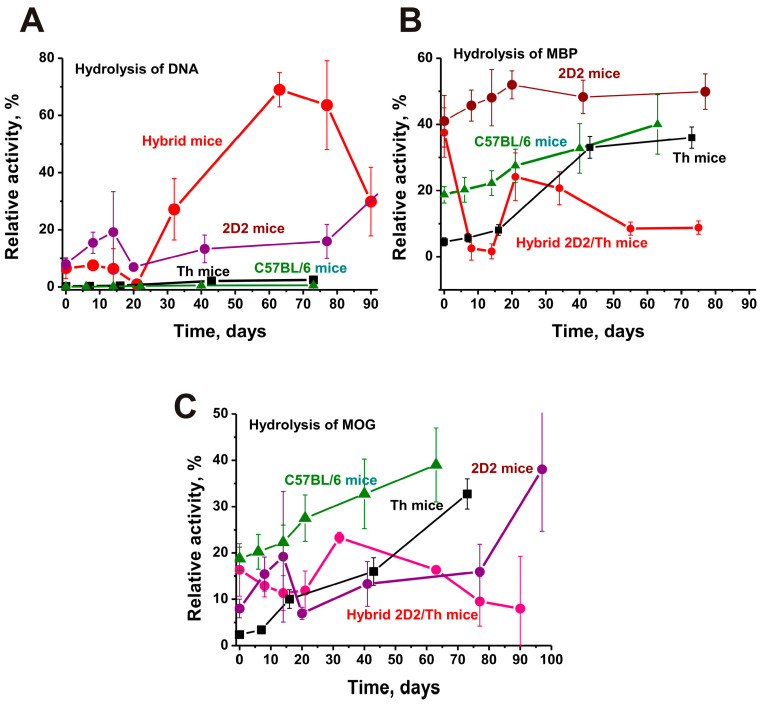
Changes over time in DNase activity of IgG-abzymes (**A**), as well as MBP- (**B**) and MOG-hydrolyzing activities (**C**). Dependencies corresponding to hybrid 2D2/Th, 2D2, Th, and C57BL/6 mice are given in different colors. The data for Th, 2D2, and C57BL/6 mice are given for comparison from our previously published articles [29,30,31,32].

**Figure 8 ijms-25-09900-f008:**
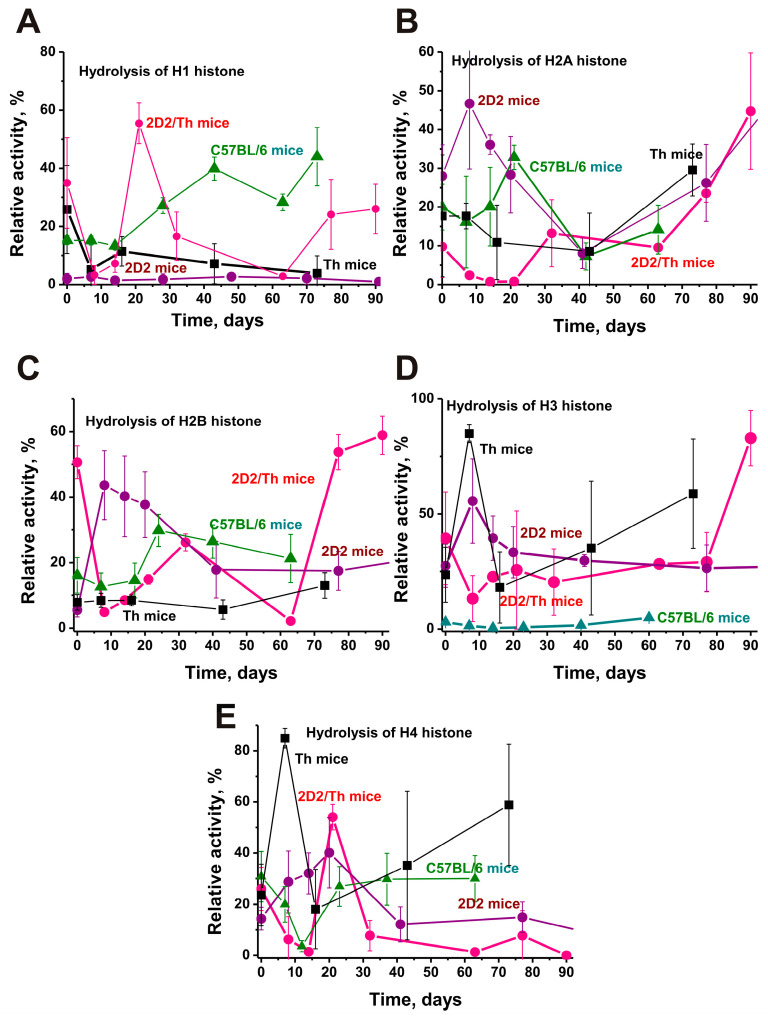
Changes over time in protease activities of IgG-abzymes hydrolyzing H1 (**A**), H2A (**B**), H2B (**C**), H3 (**D**), and H4 (**E**) histones. Dependencies corresponding to hybrid 2D2/Th, 2D2, Th, and C57BL/6 mice are given in different colors. The data for Th, 2D2, and C57BL/6 mice are given for comparison from our previously published articles [29,30,31,32]. The difference between the data of all curves is characterized by *p* < 0.05.

**Figure 9 ijms-25-09900-f009:**
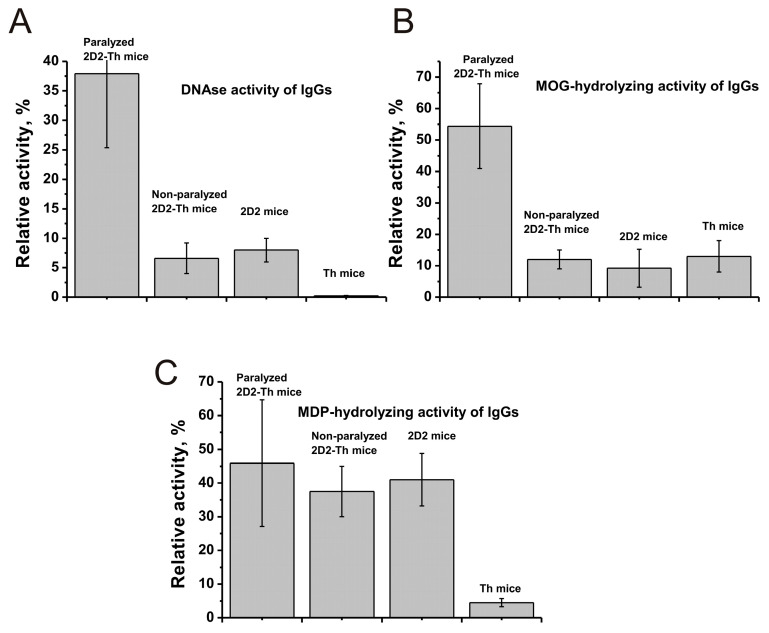
The comparison of relative DNase (**A**), MOG- (**B**), and MBP-hydrolyzing (**C**) activities of 3-month-old 2D2/Th paralyzed mice with non-paralyzed 2D2/Th, 2D2, and Th mice. The differences between the data of all curves (*p*) are given in the terx.

## Data Availability

The data that support the results of this study are included in the article and its Appendix A.

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
