# Peer review of "Cellular and Immunological Analysis of 2D2/Th Hybrid Mice Prone to Experimental Autoimmune Encephalomyelitis in Comparison with 2D2 and Th Lines"

_ijms, 2024, doi:10.3390/ijms25189900_

Round 1

Reviewer 1 Report

Comments and Suggestions for Authors

The manuscript entitled “Cellular and immunological analysis of prone to experimental autoimmune encephalomyelitis 2D2/Th hybrids in comparison with 2D2 and Th lines of mice” describes cellular, antibody and enzymatic antibodies in different spontaneous models of EAE. The data are of potential interest but the manuscript currently is difficult to follow, the results are overly long with interpretation throughout, and therefore the discussion does not read differently from the results. Moreover, the authors give the incorrect impression that EAE happens spontaneously in C57BL/6 mice throughout the manuscript; if this happens at all without MOG, this should be clarified and cited.

All of the keywords are actually phrases; these should be more succinct.

In the abstract it the use of C57BL/6 in line 2 makes it sound like these mice spontaneously develop EAE, when this is really the strain in which Th, 2D2, and 2D2xTh are made. Again this is confusing in the introduction in which the spontaneous models of EAE are mentioned; the C57BL/6 model is not spontaneous – it is induced with MOG as citations 29-31 indicate. This should be clarified. Section 2.1 also refers to C57BL/6 as a spontaneous model.

ABZs are not even mentioned in the abstract but then are the first concepts introduced. This abrupt introduction of them needs to be clarified – perhaps even just using the first sentence in the second paragraph to introduce the reader to this idea, then defining them, and discussing their prevalence in autoimmune diseases.

The introduction needs to be reorganized; there are several places in which a single sentence or two makes up a paragraph. Some of these sentences could either be combined or expanded upon to constitute 2-3 full paragraphs to make up the introduction. The authors should rewrite the introduction with the order of results in mind and be sure to introduce all concepts needed by the reader to appreciate these data.

The use of AIDs to abbreviate for autoimmune diseases is confusing since it is more commonly used for acquired immunodeficiency syndrome. Perhaps ADs can be used or not abbreviated at all.

The information in section 2.1 of the results could be incorporated into the introduction. Are there any results presented in section 2.1?

Why weren’t C57BL/6 mice used as a control for results in figures 1, 3 and 4?

Figure 2, 3 and 4 results description is too long and contains too much interpretation. The description of the results should be succinct.

Did the authors expect more colonies in the BM assays in figure 2? That was not always the case; if this was expected, it should be discussed.

Why were previous data used for the antibody analysis in some figures?

Were all mice of the same age or as it different depending on when EAE materialized (i.e., were all endpoints evaluated at 3 months)? What endpoint did they use to determine that the mice were undergoing disease? What sex of animals was used and how many per experiment?

I do not see where the supplementary data are provided; in the supplement, it looks like it is just the methods and these data are not described in the results.

Overall the discussion is very results heavy and has overlap with the current results section, which is interpretation heavy.

Minor

Word missing in the first sentence of the last paragraph of introduction.

4.1 imbred should be inbred

Comments on the Quality of English Language

Minor edits as suggested above. Main issue is the organization of the various sections.

Author Response

The manuscript entitled “Cellular and immunological analysis of prone to experimental autoimmune encephalomyelitis 2D2/Th hybrids in comparison with 2D2 and Th lines of mice” describes cellular, antibody and enzymatic antibodies in different spontaneous models of EAE. The data are of potential interest but the manuscript currently is difficult to follow, the results are overly long with interpretation throughout, and therefore the discussion does not read differently from the results. Moreover, the authors give the incorrect impression that EAE happens spontaneously in C57BL/6 mice throughout the manuscript; if this happens at all without MOG, this should be clarified and cited.

ANSWER: Formally, the C57BL/6 mouse model does not belong to the mice with spontaneous development of EAE. However, according to our data from the analysis of changes in the differentiation profile of bone marrow stem cells, the cellular and immunological analysis, these mice still experience a very-very  slow development of EAE. Taking into account your comment, we have removed the data on the spontaneous development of EAE in C57BL/6 mice.

All of the keywords are actually phrases; these should be more succinct.

ANSWER: It was corrected

In the abstract it the use of C57BL/6 in line 2 makes it sound like these mice spontaneously develop EAE, when this is really the strain in which Th, 2D2, and 2D2xTh are made. Again this is confusing in the introduction in which the spontaneous models of EAE are mentioned; the C57BL/6 model is not spontaneous – it is induced with MOG as citations 29-31 indicate. This should be clarified. Section 2.1 also refers to C57BL/6 as a spontaneous model.

ANSWER: Formally, the C57BL/6 mouse model does not belong to the mice with spontaneous development of EAE. However, according to our data from the analysis of changes in the differentiation profile of bone marrow stem cells, the cellular and immunological analysis, these mice still experience a very-very  slow development of autoimmune reactions indexes similar to those in the development of EAE after immunization of mice with  MOG. •• Before immunization of C57BL/6 mice with MOG, changes in the differentiation profiles occur similar to those after immunization and abzymes are produced. As we have shown earlier in control BALB/c and (CBAxC57BL)F1 mice not predisposed to autoimmune diseases, there is no noticeable change in the differentiation profile and abzyme production over time up to a year of life. However, in C57BL/6 mice, compared to control non-autoimmune mice, EAE develops very slowly. Taking into account your comment, we have removed the data on the spontaneous development of EAE in C57BL/6 mice.

ABZs are not even mentioned in the abstract but then are the first concepts introduced. This abrupt introduction of them needs to be clarified – perhaps even just using the first sentence in the second paragraph to introduce the reader to this idea, then defining them, and discussing their prevalence in autoimmune diseases.

ANSWER:

  • Sorry, but there was a phrase in the abstract

These changes in all strains of mice lead to the production of antibodies-abzymes against self-antigens: myelin oligodendrocyte glycoprotein (MOG), DNA, myelin basic protein (MBP), and five histones (H1-H4). The patterns of changes in the concentration of antibodies and the relative activity of abzymes during the spontaneous development of EAE in the hydrolysis of these immunogens are significantly or radically different for the three lines of mice: Th, 2D2, and 2D2/Th.

After corrections

These changes in all strains of mice lead to the production of antibodies with catalytic activities (abzymes) against self-antigens: myelin oligodendrocyte glycoprotein (MOG), DNA, myelin basic protein (MBP), and five histones (H1-H4) hydrolyzing these antigens. The patterns of changes in the concentration of antibodies and the relative activity of abzymes during the spontaneous development of EAE in the hydrolysis of these immunogens are significantly or radically different for the three lines of mice: Th, 2D2, and 2D2/Th.

The introduction needs to be reorganized; there are several places in which a single sentence or two makes up a paragraph. Some of these sentences could either be combined or expanded upon to constitute 2-3 full paragraphs to make up the introduction. The authors should rewrite the introduction with the order of results in mind and be sure to introduce all concepts needed by the reader to appreciate these data.

ANSWER:

An attempt was made to shorten the sentences. Sorry, but this resulted in a more complicated text and tautology. It was not possible to change the text significantly.

The use of AIDs to abbreviate for autoimmune diseases is confusing since it is more commonly used for acquired immunodeficiency syndrome. Perhaps ADs can be used or not abbreviated at all.

ANSWER:

  • This has been corrected with the change from autoimmune diseases (AIDs) to autoimmune pathologies (AIPs)

The information in section 2.1 of the results could be incorporated into the introduction. Are there any results presented in section 2.1?

ANSWER:

Sorry, but the justification for the choice of the model still belongs to the subsequent description of the results. Moving this part to the introduction will complicate the connection between the task set in the work and the justification for the choice of the research model. From our point of view, this part should still be in 2.1. paragraph. Reviewers of our previous publications have pointed out that the rationale for the choice of model should be included in the results.

However, one of the referees suggested moving paragraph 2.1 to “Materials and methods”. From our point of view, such a move is more justified. We moved paragraph 2.1 to “Materials and methods”.

Why weren’t C57BL/6 mice used as a control for results in figures 1, 3 and 4?

ANSWER:

The proteinuria data of C57BL/6 mice have been added to Figure 1. When we several years ago conducted the C57BL/6 mice study, we only analyzed the total lymphocytes without a specific analysis of B and T cells, so they are  absent in Figures 3 and 4.

Figure 2, 3 and 4 results description is too long and contains too much interpretation. The description of the results should be succinct.

ANSWER:

Sorry, we have already published several papers on the analysis of the differentiation profile and abzymes. In cases where the description of the results was shortened, most reviewers asked for a more detailed description. Not everyone wants to estimate by eye, looking at the pictures, which parameter is greater or less than another. Based on our experience in responding to reviewers' comments, we have come to a more optimal description of the results, which most often satisfies the majority of reviewers.

Did the authors expect more colonies in the BM assays in figure 2? That was not always the case; if this was expected, it should be discussed.

ANSWER:

This is a very difficult question. We did not expect such a significant difference in the number of colonies in the four strains of mice. However, at present there is no reasonable explanation for such a difference, and we do not want to speculate. Perhaps it will be possible to explain such a difference later.

Why were previous data used for the antibody analysis in some figures?

 ANSWER:

All mouse lines show changes in the bone marrow stem cell differentiation profile, leading to the production of abzymes that hydrolyze MBP, MOG, and DNA, which are absent in the case of non-autoimmune mice. However, the differentiation profiles in different mouse lines differ, which is reflected in the synthesis of antibodies with different relative activities. To understand the possible differences in the development of EAE in the case of different mice, we compared various parameters, including proteinuria, differentiation profiles, the relative number of B and T cells, antibody titers, and the relative activity of abzymes in the hydrolysis of various autoantigens. Ultimately, all three mouse lines – Th, 2D2, and hybrids – develop EAE, but all the parameters we studied are remarkably different. It turns out that, despite some differences, the general indicators of EAE development are changes in the stem cell differentiation profile and the production of abzymes that are harmful to the health of mice.

Were all mice of the same age or as it different depending on when EAE materialized (i.e., were all endpoints evaluated at 3 months)? What endpoint did they use to determine that the mice were undergoing disease? What sex of animals was used and how many per experiment?

ANSWER We used three-month-old male mice born at approximately the same time with a difference of 1-3 days (a total of 70 mice). From the 70 mice, groups of 7 mice were formed. Decapitation of 7 mice was carried out at 3 months of their life (zero time), the next group of 7 mice 8 at days after zero time and then after 14, 21, 32, 63, 77, and 90 days after zero time, as shown in the figures.

 I do not see where the supplementary data are provided; in the supplement, it looks like it is just the methods and these data are not described in the results.

ANSWER Since in previous works we used the same methods as in this study, their repeated presentation leads to repetition of what has already been published and plagiarism. Taking this into account, some of the previously described methods (analysis of B and T lymphocytes)  are given in the appendix

Overall the discussion is very results heavy and has overlap with the current results section, which is interpretation heavy.

ANSWER For many specialists, including immunologists, the data on the analysis of the differentiation profile of stem cells and the relationship between differentiation and the production of catalytic antibodies are perceived as very difficult. Based on our experience in writing previous articles, we came to this version of the description of the results and discussion. In addition, the Discussion provides data of other types of information and from other articles to explain the results obtained.

 Minor

Word missing in the first sentence of the last paragraph of introduction.

4.1 imbred should be inbred

Thank you for your very helpful comments

Best regards

Professor

Georgy A. Nevinsky

Reviewer 2 Report

Comments and Suggestions for Authors

Major:

1. Include EAE score pattern of 3 lines of mouse 

Minor:

1. Discuss genotypic and phenotypic  difference between 3 lines of mouse 

2.  2D2 = Homo, 2D2/Th= Hetero, TH= Wild ??

3. Discuss EAE pattern and demyelination, inflammation?

Author Response

Major:

  1. Include EAE score pattern of 3 lines of mouse 

Minor:

  1. Discuss genotypic and phenotypic  difference between 3 lines of mouse 
  2. 2D2 = Homo, 2D2/Th= Hetero, TH= Wild ??
  3. Discuss EAE pattern and demyelination, inflammation?

Answer:

Sorry, but we did not understand the meaning of your comments. In our work, we did not obtain new Th, 2D2 mouse lines, we used ready-made mouse lines. All data on these mouse lines, Including Th/2D2 hybrids, have already been described in the literature and in the introduction are given references  [24-26,28-32, 38,39]. In our experiments, we did not find any noticeable differences in mice from their characteristics described in the literature. Considering these are the indicators that you ask to provide, there was no point in studying them in this work. This information has been added to the materials and methods with links to literary data on all three types of mice.

Below I additionally provide literary data on the characteristics of these mice.

Individuals of the Th line carry a transgenic heavy chain of immunoglobulins IgH specific to the MOG protein, which is part of both immunoglobulin molecules and receptors on the surface of B-lymphocytes (BCR) (https://pubmed.ncbi.nlm.nih.gov/9653093/). About 30% of B-lymphocytes in this model carry a transgenic B-cell receptor, and the blood of such mice has a high level of IgM, IgG2a, and IgG2b antibodies to recombinant MOG (1-125). According to the literature, individuals of this line do not show signs of paralysis throughout their lives. When immunized with MOG35-55, Th mice develop a more severe disease than wild-type individuals (C57BL/6).

Mice of the 2D2 line carry a transgenic T-lymphocyte receptor (TCR) specific to the MOG35-55 peptide with rearranged V(D)J segments of the α and β chains of the receptor. These DNA sequences were obtained from T-lymphocytes of mice with EAE induced by the MOG35-55 peptide and were introduced as part of plasmids into the pronucleus of mouse oocytes to create a transgenic line (https://pubmed.ncbi.nlm.nih.gov/12732654/). According to the literature, 4-6% of individuals of the 2D2 line are characterized by spontaneous development of experimental encephalomyelitis with paralysis of the limbs, inflammation in the spinal cord and demyelination of axons.

Hybrid individuals obtained by crossing these lines (Th and 2D2) carry both a MOG-specific B-cell receptor on the surface of B-lymphocytes and a MOG35-55 peptide-specific T-cell receptor on the surface of T-lymphocytes. Mice of this line are characterized by rapid spontaneous development of EAE with limb paralysis, spinal cord inflammation, and axonal demyelination (46-51% of individuals) (https://pubmed.ncbi.nlm.nih.gov/16955141/, https://pubmed.ncbi.nlm.nih.gov/16955140/).

C57BL/6 are a control line of mice that are not susceptible to paralysis. The transgenic Th line, created on the genetic basis of the C57BL/6 line, also does not typically show signs of paralysis, while according to the literature, this manifestation of the disease is typical for 4-6% of individuals of the 2D2 line (https://pubmed.ncbi.nlm.nih.gov/9653093/, https://pubmed.ncbi.nlm.nih.gov/12732654/). In hybrid individuals of the 2D2 and Th lines, symptoms of EAE with paralysis of the tail and limbs, according to literary data, appear in 46-51% of individuals on average from the 6th week (https://pubmed.ncbi.nlm.nih.gov/16955141/, https://pubmed.ncbi.nlm.nih.gov/16955140/

other details about these mouse lines can also be found in the links provided

Best regards

Professor

Georgy A. Nevinsky

Reviewer 3 Report

Comments and Suggestions for Authors

I have reviewed the paper by Aulova et al.

The data is solid but the narrative of the paper needs extensive revision and English editing.

For example “As shown earlier [24-26,28-32], some antibodies against DNA have DNase activity.

should be “As it has been shown previously, some antibodies against DNA have DNase activity [24-26, 28-32]”

There is an excessive number of citations in the Introduction. They should keep it in focus with the specific topic of the study and remove unnecessary information.

2.1 should be in Material and methods and not in Results.

Information in 2.2 should be in Material and methods and in Discussion, and keep findings on Figure 1 in Results.

There is an overall lack of statistical analysis from the graphs, despite authors talking about “significantly lower”

Graphs appear too small and difficult to visualize. It may be a good idea to put C57BL/6 data in a separate set of graphs, and these curves are rather disruptive in visualizing the change in  2D2/Th, and the parent controls.

Narrative of results needs a better paragraph regrouping and needs to be more concise.

Comments on the Quality of English Language

Needs extensive editing.

Author Response

I have reviewed the paper by Aulova et al.

The data is solid but the narrative of the paper needs extensive revision and English editing.

For example “As shown earlier [24-26,28-32], some antibodies against DNA have DNase activity.”should be “As it has been shown previously, some antibodies against DNA have DNase activity [24-26, 28-32]”

Answer: It was corrected

There is an excessive number of citations in the Introduction. They should keep it in focus with the specific topic of the study and remove unnecessary information.

Answer:

Catalytic antibodies are a relatively new direction in immunology, with which even immunologist  are little familiar. To understand the overall picture in the research of catalytic antibodies, we have included references related to this topic. From our point of view, these references help to understand the overall situation in this area of

research.

2.1 should be in Material and methods and not in Results.

Answer

It was done – section 2.1 was moved to Material and methods

Information in 2.2 should be in Material and methods and in Discussion, and keep findings on Figure 1 in Results.

Answer:

It was done

There is an overall lack of statistical analysis from the graphs, despite authors talking about “significantly lower”

Answer:

All graphs show average data for 7 mice and standard deviations, and p values are given in the text.

Graphs appear too small and difficult to visualize. It may be a good idea to put C57BL/6 data in a separate set of graphs, and these curves are rather disruptive in visualizing the change in  2D2/Th, and the parent controls.

Answer

All graphs have been enlarged where possible. Showing mouse data in separate figures for C57BL/6 would make it difficult to compare results across the four mouse strains.

Narrative of results needs a better paragraph regrouping and needs to be more concise.

Answer

Sorry, your comment is very vague.The data in the article seems difficult to understand for many scientists. We are not sure that a more concise description of the results will make them understandable, sorry

Thank you for your very helpful comments

Best regards

Professor

Georgy A. Nevinsky

Round 2

Reviewer 1 Report

Comments and Suggestions for Authors

Thank you for the responses to the reviewer's concerns. Some issues remain.

The concern regarding keywords was not addressed sufficiently. The problem is that there are no words, only phrases. I think these could be changed.

The concern regarding paragraphs was not addressed sufficiently. For instance, the first three paragraphs in the introduction can be combined (lines 47-58 can be one paragraph). They are all referring to ABZs. Then the current paragraphs 4 and 5 can be combined since they have introduced AIPs generally then move to EAE; the following EAE paragraph can be combined (lines 59-68 can be one paragraph).

Is the conclusion paragraph in the correct place? It seems like it should be the last paragraph of the discussion.

Minor

Line 49 should be “Natural abzymes from biological liquids of mammals are well described in [9-26].”

Comments on the Quality of English Language

See comments above regarding paragraphs in the introduction.

Author Response

Thank you for the responses to the reviewer's concerns. Some issues remain.

The concern regarding keywords was not addressed sufficiently. The problem is that there are no words, only phrases. I think these could be changed.

 Answer

Sorry, tried to shorten the code words, but further shortening results in the absence of the content of the article in these words

The concern regarding paragraphs was not addressed sufficiently. For instance, the first three paragraphs in the introduction can be combined (lines 47-58 can be one paragraph). They are all referring to ABZs. Then the current paragraphs 4 and 5 can be combined since they have introduced AIPs generally then move to EAE; the following EAE paragraph can be combined (lines 59-68 can be one paragraph).

Answer

It was done

Is the conclusion paragraph in the correct place? It seems like it should be the last paragraph of the discussion.

Answer

Sorry, but according to the rules of  this journal the conclusion should be after the Materials and methods.

Minor

Line 49 should be “Natural abzymes from biological liquids of mammals are well described in [9-26].”

Answer

It was corrected

Thanks a lot for useful comments

Sincerely

 Prof. Georgy A. Nevinsky

Reviewer 2 Report

Comments and Suggestions for Authors

Good

Author Response

Thak you very much

Georgy Nevinsky

Reviewer 3 Report

Comments and Suggestions for Authors

I. Authors claim that "Catalytic antibodies are a relatively new direction in immunology" and that "To understand the overall picture in the research of catalytic antibodies, we have included references related to this topic. From our point of view, these references help to understand the overall situation in this area of research". How is putting 26 references helping this at all?? "The literature describes artificial сatalytically active antibodies-abzymes (ABZs) 47 against stable analogs of transition states of different chemical reactions and their use [1-48 8]. Natural abzymes from biological liquids of mammals well described in [9-26]. "

Please retain only a few significant references to make a point.

II. Statistical significant findings need to pointed In The Graphs.

III. Graphs are still not large enough for proper visualization (e.g. Figure 7 is impossible to be read). All figures need to enlarged (there is plenty of blank space around) as this will certainly have a negative impact on the readers.

IV. Authors respond that "Showing mouse data in separate figures for C57BL/6 would make it difficult to compare results across the four mouse strains." nevertheless, no C57 curves are shown for Figures 3 and 4.

V. An example of information that should not be in Results but in Discussion is "The fact that 2D2/Th demonstrates an increased concentration of antibodies against 278 DNA already in 3-month-old mice and the development of EAE within 90 days may play 279 an important role in the accelerated development of this pathology in 2D2/Th mice. As it 280 has been shown previously, some antibodies against DNA have DNase activity [24-26,28-281 32]. DNA-hydrolyzing abzymes penetrate the membranes of cells and nuclei and hydro-282 lyze chromatin DNA, stimulating cell apoptosis [43,44]. This leads to an increase in the 283 concentration of DNA complexes with histones in the blood. However, DNA complexes 284 with histones are the main antigens in the formation of anti-DNA antibodies [45]. An in-285 crease in the concentration of such antibodies leads to increased cell apoptosis and the 286 flare-up of autoimmune processes [24-26]."

Author Response

. Authors claim that "Catalytic antibodies are a relatively new direction in immunology" and that "To understand the overall picture in the research of catalytic antibodies, we have included references related to this topic. From our point of view, these references help to understand the overall situation in this area of research". How is putting 26 references helping this at all?? "The literature describes artificial сatalytically active antibodies-abzymes (ABZs) 47 against stable analogs of transition states of different chemical reactions and their use [1-48 8]. Natural abzymes from biological liquids of mammals well described in [9-26]. "

Please retain only a few significant references to make a point.

Answer:

Sorry, I would gladly follow your advice and remove some of the references. But according to the new rules of this journal, the number of references to your own publications should not exceed 15%. In order to include all the references to your own works, which are necessary for understanding the results of this study, I must provide at least 8 references (there are many more, but there is no possibility to add more references). All the works that I could refer to and which are related to catalytic antibodies, I have provided in front. Unfortunately, I cannot remove any of them. If I remove even one, the number of references to my own publications will exceed 15%, and as a result, the article will not be accepted by the journal for publication. Sorry, but I have no other choice - I am forced to leave all the references.

  1. Statistical significant findings need to pointed In The Graphs.

Answer:

There are graphs on which it is easy to indicate Statistical significant. Where and how to put Statistical significant on our graphs I do not understand. Taking into account your comment, I added this information to the captions to the figures. All figures are enlarged

III. Graphs are still not large enough for proper visualization (e.g. Figure 7 is impossible to be read). All figures need to enlarged (there is plenty of blank space around) as this will certainly have a negative impact on the readers.

  1. Authors respond that "Showing mouse data in separate figures for C57BL/6 would make it difficult to compare results across the four mouse strains." nevertheless, no C57 curves are shown for Figures 3 and 4.

Answer:

When we several years ago conducted the C57BL/6 mice study, we only analyzed the total lymphocytes without a specific analysis of B and T cells, so they are  absent in Figures 3 and 4.

  1. An example of information that should not be in Results but in Discussion is "The fact that 2D2/Th demonstrates an increased concentration of antibodies against 278 DNA already in 3-month-old mice and the development of EAE within 90 days may play 279 an important role in the accelerated development of this pathology in 2D2/Th mice. As it 280 has been shown previously, some antibodies against DNA have DNase activity [24-26,28-281 32]. DNA-hydrolyzing abzymes penetrate the membranes of cells and nuclei and hydro-282 lyze chromatin DNA, stimulating cell apoptosis [43,44]. This leads to an increase in the 283 concentration of DNA complexes with histones in the blood. However, DNA complexes 284 with histones are the main antigens in the formation of anti-DNA antibodies [45]. An in-285 crease in the concentration of such antibodies leads to increased cell apoptosis and the 286 flare-up of autoimmune processes [24-26]."

Answer:

Sorry, but this remark is not entirely clear to us. These data, from our point of view, should be in the results section - since these are the results

In the discussion, these results are also discussed, but in more detail, with the use of additional literary data.

Thanks a lot for useful comments

Sincerely

 Prof. Georgy A. Nevinsky

Round 3

Reviewer 3 Report

Comments and Suggestions for Authors

Authors appear unwilling to make the necessary correction in a revised version.

Therefore, I cannot recommend acceptance of this manuscript.

Author Response

I have not answer 

In addition

Please forgive me, but the third reviewer's remark did not concern our self-citation links, but a large number of links to other people's works. But from our point of view, all of them are necessary for understanding the general situation in this area of research.

In total, the article contains 53 links and to to our works 8 references – 15 %. Please forgive me, but unfortunately it is not possible to do what you are asking.  To understand what is written in this article, it is necessary to provide all three 24-26 links. The same applies to links 30-32. Without these references, the meaning of this article will not be clear - it will hang in the air, sorry. Sorry, but the number of references to own works meets the requirements of the journal - 15%

With best wishes